# DeepSeek-AI-enhanced virtual reality training for mass casualty management: Leveraging machine learning for personalized instructional optimization

Zhe Li, Lei Shi, Mingyu Pei, Wan Chen, Yutao Tang, Guozheng Qiu, Xibin Xu, Liwen Lyu*

Department of Emergency, The People's Hospital of Guangxi Zhuang Autonomous Region, Guangxi Academy of Medical Sciences, Nanning, China

* iculvliwen@163.com

## Abstract

### Objective

This study aimed to evaluate the effectiveness of a virtual reality (VR) training system for mass casualty management, integrating artificial intelligence (AI) and machine learning (ML) to analyze trainee performance and error patterns. The goal was to identify key predictors of performance, generate personalized feedback, and provide actionable recommendations for optimizing VR-based medical training.

### Materials and methods

A total of 196 medical professionals participated in a 1-hour VR training session, followed by a 20-question assessment and a post-training evaluation survey. The DeepSeek AI framework was employed to analyze the data, utilizing clustering analysis, principal component analysis (PCA), and random forest models. Descriptive statistics, error rates, and correlation analyses were performed using R software (version 4.1.2). Machine learning models were trained to predict performance outcomes, and feature importance was assessed using the Gini index. Personalized feedback reports were generated based on clustering and error analysis results.

### Results

The study identified three distinct trainee clusters, with the highest-performing group excelling in Trauma Assessment and Clinical Case Analysis. However, high error rates were observed in Clinical Case Analysis (69.4%) and Trauma Assessment (67.3%), indicating areas for targeted improvement. Machine learning models highlighted replacing traditional teaching methods (IncNodePurity = 25.76) and stimulating learning interest (IncNodePurity = 13.08) as the most critical factors influencing

**Data availability statement:** All relevant data are within the manuscript and its Supporting information files.

**Funding:** Guangxi Medical and Health Appropriate Technology Development and Promotion Application Project [grant number: S2023013]; Guangxi Natural Science Foundation [grant number: 2024GXNSFAA010071] and Guangxi Natural Science Foundation [grant number: 2024GXNSFAA010012]. The funders had no role in study design, data collection and analysis, decision to publish, or preparation of the manuscript.

**Competing interests:** The authors have declared that no competing interests exist.

learning outcomes. AI-driven feedback provided actionable recommendations, such as redesigning complex scenarios and enhancing system usability.

## Conclusions

This study demonstrates the potential of integrating AI with VR training to create a more personalized and effective learning experience for medical professionals. The findings underscore the importance of adaptive, data-driven approaches in medical education, particularly in high-stakes environments such as emergency medicine. Future research should explore hybrid training models and incorporate physiological data to further enhance the efficacy of VR-based training systems.

## Introduction

Medical education, particularly in high-pressure fields such as emergency medicine, requires continuous innovation to enhance the effectiveness of training programs [1]. Traditional approaches, such as lectures and written assessments, while valuable, often fail to adequately prepare healthcare professionals for the dynamic and complex nature of real-world clinical practice [2]. As the demand for more efficient and effective training systems grows, virtual reality (VR) has gained recognition as an innovative approach, offering immersive, interactive, and scenario-based learning environments [3,4]. VR-based simulations have shown considerable potential in enhancing clinical decision-making, skill acquisition, and overall performance, particularly in mass casualty management and emergency care settings [5].

Despite the increasing integration of VR in medical training, several challenges remain in fully optimizing its effectiveness. Although VR training is beneficial for fostering realistic practice, it often lacks personalized feedback, dynamic adaptation to trainee performance, and the ability to adjust learning pathways according to individual needs [6]. These limitations hinder the ability to offer tailored learning experiences that effectively address the diverse requirements of trainees. Research has highlighted these issues, with studies indicating that VR systems typically offer static scenarios that do not adapt in real time to the learner's progress or specific requirements. For example, while VR provides immersive training, the absence of individualized feedback limits its potential to enhance learner engagement and performance [7].

To overcome these limitations, leveraging artificial intelligence (AI) and machine learning (ML) offers a promising pathway to enhance the adaptability and personalization of VR-based training. A theoretical foundation is essential for guiding such integration. Kolb's Experiential Learning Theory, which outlines four stages—concrete experience, reflective observation, abstract conceptualization, and active experimentation—offers a meaningful lens through which to understand immersive medical training [8]. Within VR environments, leaners engaged directly with high-fidelity clinical scenarios as concrete experiences. AI-driven feedback mechanisms can facilitate reflective observation by highlighting critical actions or missed steps, while pattern recognition

algorithms assist with abstract conceptualization by identifying trends in learner behavior. Finally, repeated practice with system-guided prompts supports active experimentation, allowing learners to iteratively refine their skills. Complementing this model, learning retention frameworks such as the Ebbinghaus forgetting curve emphasize the importance of timely, targeted reinforcement [9]. AI systems are particularly well-suited to monitor individual learning trajectories and deliver personalized prompts to revisit or reinforce difficult concepts, thereby supporting long-term retention and performance improvement.

DeepSeek [10,11], an advanced AI-powered analytical framework, is designed to operationalize these pedagogical principles. By analyzing interaction data generated throughout VR sessions, DeepSeek identifies learning gaps, tracks behavioral patterns, and provides real-time feedback tailored to each learner's evolving needs. In this way, it supports both the experiential learning cycle and memory retention strategies, enabling a more effective, individualized training process. This study integrates DeepSeek into a VR-based mass casualty training system to explore how personalized, adaptive instruction can enhance the preparedness of medical professionals in emergency settings.

The present study specifically investigates the effectiveness of a VR-based training system for mass casualty treatment among medical professionals, incorporating AI-driven feedback and machine learning models to optimize performance and learning outcomes. Through this innovative approach, we aim to explore how the integration of VR, AI, and machine learning can create a more efficient and personalized medical training experience, addressing the evolving needs of healthcare professionals in real-world clinical environments.

## Materials and methods

### Study design, setting, and ethics

This study employed a retrospective design to assess the effectiveness of a virtual reality (VR) training system for mass casualty treatment among medical staff. It was conducted at the Emergency Department of Guangxi Zhuang Autonomous Region People's Hospital in Nanning, China, from January to December 2024. All participants provided written informed consent and completed a standardized assessment following the VR training. The study was approved by the Ethics Committee of Guangxi Zhuang Autonomous Region People's Hospital (Approval No. KY-SY-2023-014), and all personal data were anonymized to ensure participant confidentiality.

### Participants

A total of 196 trainees participated in the study, consisting of emergency physicians, resident trainees, and nurses. These participants were selected based on the following inclusion criteria: (1) willingness to participate and provide informed consent, and (2) ability to complete the full training program and assessments. Exclusion criteria included: (1) inability to fully complete the training or assessments, and (2) refusal to participate. All trainees underwent the same training and evaluation process, and no separate grouping was conducted for comparison. Additionally, to facilitate comparative analysis, we extracted archival data from a cohort of 196 trainees who underwent conventional, lecture-based emergency training between 2021 and 2023. These learners completed the same theoretical assessment, although instructional formats, training durations, and specific contents were not fully standardized during that period due to ongoing curriculum adjustments.

### VR training system

The VR training system used in this study was developed in-house by the Emergency Department of Guangxi Zhuang Autonomous Region People's Hospital. The software, named Road Traffic Injury VR Software 1.0, was specifically designed to simulate road traffic injury scenarios and mass casualty situations. Using the HTC VIVE VR platform, the system provided an immersive environment where participants could engage in realistic simulations involving injury assessment, triage, trauma management, and transport decision-making.

The training session lasted for 1 hour, during which participants interacted with the VR environment, performing key tasks related to on-site evaluation, injury classification, trauma assessment, and transport decisions. While the session was not formally assessed, the knowledge and skills addressed during the training were closely aligned with the content of the subsequent theoretical exam, ensuring that the training directly supported the learning objectives for the final evaluation.

## Questionnaire design

Data were gathered from two primary sources. After the training session, participants completed a 20-question multiple-choice assessment through a QR code-based online platform. This assessment tested knowledge and decision-making skills related to the training content. Additionally, participants completed a post-training survey evaluating their VR experience, which included questions about the realism of the simulation, usability of the system, physical comfort while using the VR equipment, learning effectiveness, and overall engagement.

## Data collection and analytical methods

The data were analyzed using R software (version 4.1.2) and the DeepSeek AI framework. Descriptive statistics, such as means, standard deviations, and frequency distributions, were computed for demographic variables, performance indicators, and evaluation scores. Error rates were computed for each question and performance dimension to identify areas where trainees struggled, and high-error questions were flagged for further analysis.

Clustering analysis was performed using Principal Component Analysis (PCA) and k-means clustering. PCA, implemented through the factoextra package, was used to reduce dimensionality and visualize patterns in trainee performance and evaluation responses. k-means clustering, performed using the stats package, grouped participants based on their performance and evaluation responses, with the optimal number of clusters determined through the elbow method. Data visualizations were generated using the ggplot2 package. Correlation matrices were visualized using the corrplot package.

To predict performance outcomes based on demographic data and evaluation scores, Random Forest (RF) models were trained using the randomForest package. Random Forest is an ensemble learning method that constructs multiple decision trees and combines their outputs for classification or regression tasks. In this study, RF was used to assess how demographic factors (e.g., age, prior experience) and evaluation scores (e.g., task accuracy, response time) correlate with overall performance. Model performance was assessed using 10-fold cross-validation, a technique to reduce overfitting and ensure robust model evaluation. Feature importance was calculated using the Gini index, which indicates the contribution of each feature (variable) to the model's predictive power.

The DeepSeek AI framework was employed to analyze the data and identify key predictors of performance, offering personalized feedback to trainees. DeepSeek's AI capabilities allowed for a more nuanced understanding of what factors (e.g., specific performance metrics or areas of difficulty) influenced trainee success. The framework also assessed how replacing traditional teaching methods with immersive VR training influenced learning outcomes and trainee engagement.

Statistical analysis was conducted to assess the relationships between evaluation items and performance metrics. The distribution of data was first assessed, and appropriate statistical tests were applied. For continuous variables with normal distribution, t-tests were used to compare group means, while Mann-Whitney U-tests were employed for non-normally distributed data. To explore the associations between evaluation items and performance scores, Pearson's correlation coefficient was calculated. All statistical analyses were performed using R software (version 4.1.2), with significance set at $P < 0.05$.

## AI-driven feedback and teaching optimization

The DeepSeek framework played a pivotal role in transforming raw data into actionable feedback and teaching recommendations. Through error analysis and clustering results, the framework identified common challenges, such as difficulties in Clinical Case Analysis and Trauma Assessment. Based on individual performance, DeepSeek generated

personalized feedback for each trainee, providing tailored recommendations for improvement in areas with high error rates.

DeepSeek also provided recommendations for optimizing the VR training system. This included redesigning complex scenarios to address high-error dimensions, improving system usability and realism to enhance engagement, and developing adaptive learning paths for different trainee clusters to personalize the training experience.

## Results

### Trainees characteristics

As shown in Table 1, a total of 196 trainees participated in this study (Fig 1). Regarding gender, 108 trainees (55.1%) were male and 88 (44.9%) were female. The mean age of the trainees was 26.23±2.18 years. Among the trainees, 78 (39.8%) were further – study students, 104 (54.06%) were residency trainees, and 14 (7.14%) were nurses. In the theoretical examination, the mean score of the trainees was 60.63±24.68. The average time taken to complete the theoretical examination was 501.88±325.67 seconds. These baseline characteristics can serve as a basis for further exploring the performance differences among different groups of trainees and analyzing the influencing factors related to the examination results.

### Comparison with historical cohort under traditional training

To provide a contextual benchmark for the observed learning performance, we compared the VR+AI training group with a historical cohort (n = 196) who received traditional, lecture-based emergency training between 2021 and 2023.

**Table 1. Baseline characteristics of trainees.**

| Variable | Category/Statistic | Value |
|---|---|---|
| Total Trainees | N | 196 |
| Gender | Male (n,%) | 108, 55.1% |
| | Female (n,%) | 88, 44.9% |
| Age(years) | Mean±SD | 26.23±2.18 |
| Academic Type | Further Study (n,%) | 78, 39.8% |
| | Residency (n,%) | 104, 54.06% |
| | Nursing (n,%) | 14, 7.14% |
| Total Score | Mean±SD | 60.63±24.68 |
| Time Spent (seconds) | Mean±SD | 501.88±325.67 |

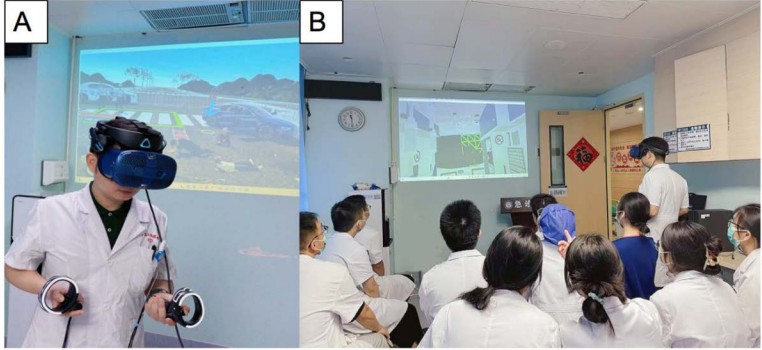

**Fig 1. The scenario in which instructors and trainees are undergoing VR - based training.**

Both groups completed the same standardized theoretical assessment. The mean total score of the VR+AI group was 60.63±24.68, while the traditional training group achieved a mean score of 55.79±17.89, as illustrated in Fig 2. The difference was statistically significant ($p < 0.05$).

## Machine learning-driven classification of trainee error patterns

As shown in Table 2, the top ten most frequently misclassified questions were primarily in Clinical Case Analysis, with Question 20 exhibiting the highest error rate (69.4%), followed by Question 19 (67.3%) and Question 18 (58.2%). Errors were also prevalent in Injury Classification, Trauma Assessment, and Batch Casualties, indicating a need for targeted instructional enhancements. The machine learning classification model demonstrated strong predictive performance, achieving an AUC of 0.976 (Fig 3), confirming its effectiveness in distinguishing trainees with varying competency levels. Furthermore, clustering analysis (Fig 4) identified three distinct trainee clusters, each with different error patterns, highlighting specific learning need, which providing valuable insights for refining VR-based training strategies.

## Principal component and clustering analysis of trainee response time and accuracy

Principal Component Analysis (PCA) was conducted to examine the relationship between trainees' response times and accuracy (Fig 5). The first two principal components (PC1 and PC2) together explained 100% of the variance, with PC1

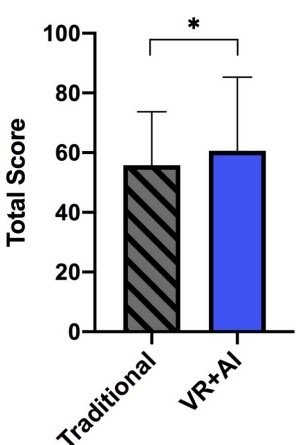

**Fig 2. Comparison of theoretical assessment scores between VR+ AI group and traditional training group.**

**Table 2. Top 10 questions with the highest error rates.**

| Question | Dimension | Error rate | Question stem |
|---|---|---|---|
| Question20 | Clinical Case Analysis | 0.694 | Female, 55 years old, was hit by a car while riding a motorcycle and rolled to the ground… |
| Question19 | Clinical Case Analysis | 0.673 | A batch of casualties was treated on site. One casualty in the red zone had a GCS score of 12... |
| Question18 | Clinical Case Analysis | 0.582 | Which of the following statements regarding initial re-evaluation is incorrect? |
| Question17 | Clinical Case Analysis | 0.566 | A green zone patient with trauma to his left upper limb suddenly fainted and fell to the ground… |
| Question6 | Injury Classification | 0.429 | As the first rescue team arriving at the scene, the first thing they do after triaging the injured is… |
| Question8 | Injury Classification | 0.429 | Emergency response early warning linkage system includes… |
| Question10 | Trauma Assessment | 0.429 | The core points of trauma reassessment are… |
| Question2 | Batch Casualties | 0.423 | The A in Initial Injury Assessment ABCDE refers to… |
| Question9 | Injury Classification | 0.393 | Recommended for on-site and 120 dispatch center information transfer… |

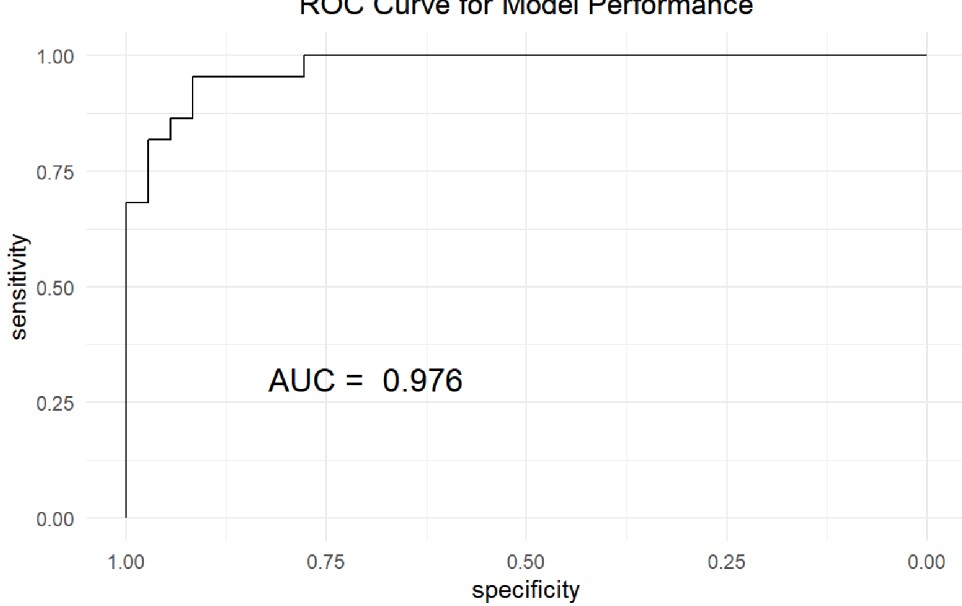

**Fig 3. ROC curve of the classification model.**

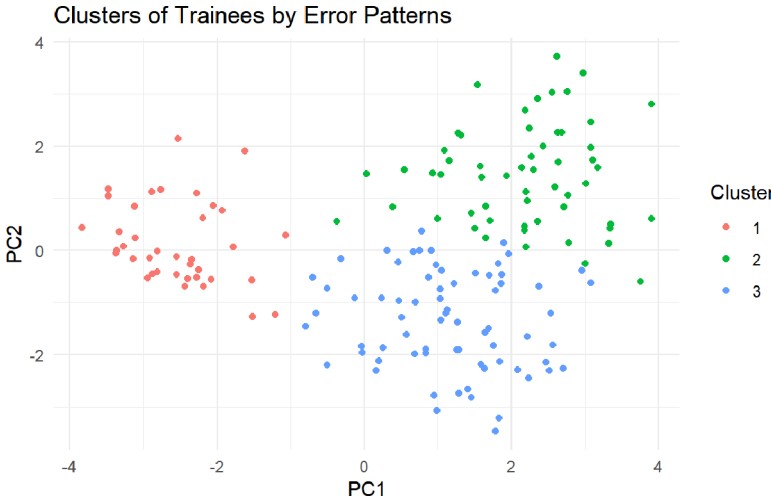

**Fig 4. PCA-based clustering of trainee error patterns.**

accounting for 57.46% and PC2 for 42.54% (Table 3). PC1, primarily influenced by TimeUsed (loading = 0.707) and TotalScore (loading = −0.707), reflects the speed-accuracy tradeoff, where trainees with longer response times tend to have lower accuracy, while those who respond faster achieve higher accuracy. PC2, contributing 42.54% of the variance, captures a general balance between response time and accuracy across trainees.

This segmentation provides actionable insights for training optimization—slower, lower-accuracy trainees may need targeted interventions, while faster but moderately accurate trainees could benefit from focused reinforcement on critical decision-making.

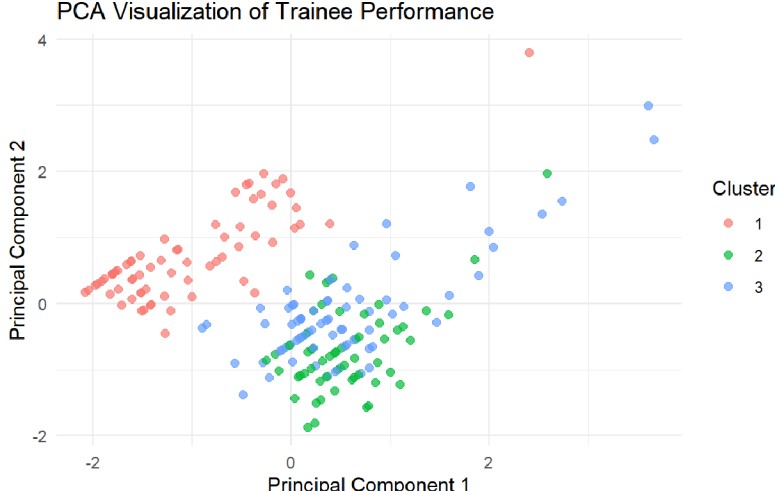

**Fig 5. PCA-based clustering of trainee performance.**

**Table 3. Principal component analysis of trainee response time and accuracy.**

| Component | Standard deviation | Proportion of variance | Cumulative variance |
|---|---|---|---|
| PC1 | 1.072 | 57.46% | 57.46% |
| PC2 | 0.922 | 42.54% | 100% |

## Unveiling trainee experience through machine learning

Machine learning analysis provided critical insights into how experience factors influence trainee performance, offering guidance for refining VR-based training. Clustering analysis identified three distinct trainee groups, with one cluster (Cluster 2) reporting high satisfaction across all evaluation metrics (e.g., Realistic Feeling = 4.93, Good Learning Effect = 4.96) and demonstrating balanced performance, underscoring the effectiveness of VR training for engaged learners. The random forest model identified Replaces Traditional Teaching (IncNodePurity = 25.76), Stimulates Interest (IncNodePurity = 13.08), and No Discomfort (IncNodePurity = 11.09) as the most influential factors in predicting TotalScore (Table 4), suggesting that enhancing engagement and minimizing discomfort can significantly improve learning outcomes (Fig 6). In contrast, Multi Dimensional Viewing exhibited a negative impact (IncNodePurity = 1.11), indicating that excessive visual complexity may hinder performance. The linear regression model highlighted Easy Interface (β = 9.88) and Convenient Equipment (β = 9.28) as positive contributors, reinforcing the importance of intuitive design and accessible technology. However, its limited explanatory power ($R^2$ = 0.07, p = 0.19) underscores the advantage of machine learning in capturing complex, non-linear relationships.

## AI-driven personalized feedback and system evaluation

**Comprehensive evaluation of the VR training system.** The analysis in Sections 3.1–3.4 provides a comprehensive evaluation of the current VR training system, highlighting its strengths and areas for improvement. The system demonstrates strong potential for enhancing mass casualty training, particularly for high-performing trainees who excel in complex scenarios such as Trauma Assessment and Clinical Case Analysis. However, the clustering and error analysis revealed critical areas for improvement, including the need for enhanced realism and usability (e.g., for Cluster 1 trainees) and targeted interventions for high-error dimensions such as Clinical Case Analysis and Trauma Assessment.

**Table 4. Variable importance in predicting totalscore: insights from linear model and random forest.**

| Variable | Estimate (Linear Model) | Importance (Random Forest) |
|---|---|---|
| Realistic Feeling | 6.70 | 3.25 |
| Simple Operation | −3.88 | 10.31 |
| Easy Interface | 9.88 | 4.92 |
| Convenient Equipment | 9.28 | 4.76 |
| Multi Dimensional Viewing | −16.61 | 1.11 |
| Stimulates Interest | −2.06 | 13.08 |
| Good Learning Effect | −5.23 | 0.00 |
| Replaces Traditional Teaching | 2.99 | 25.76 |
| No Discomfort | −5.04 | 11.09 |
| No Eye Pain | 1.97 | 10.06 |

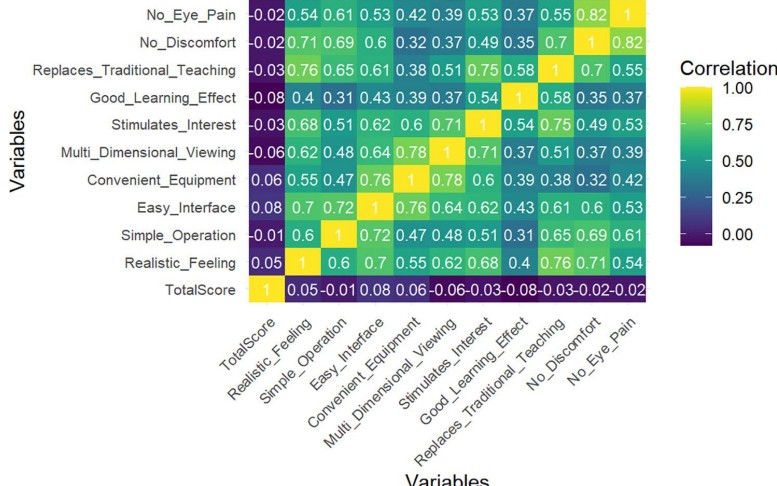

**Fig 6. Correlation heatmap of experience factors influencing trainee performance.**

**AI-driven personalized feedback reports.** To address these challenges and optimize trainee outcomes, we utilized AI-driven personalized feedback reports powered by Deepseek (Fig 7A). These reports synthesize insights from clustering, error analysis, and machine learning models to provide actionable recommendations for individual trainees (Fig 7B).

## Discussion

This study assessed the effectiveness of a virtual reality (VR) training system for mass casualty management, integrating AI-driven analysis to identify performance patterns and error rates among medical professionals. The key findings highlight significant insights into how VR training, enhanced with machine learning, can improve the quality and personalization of medical education. By leveraging AI-driven clustering analysis, trainees were grouped into three distinct clusters based on their performance, with the highest-performing group excelling in key areas such as trauma assessment and clinical case analysis. However, higher error rates were identified in clinical decision-making tasks, particularly in clinical case analysis and trauma assessment, indicating areas that require further attention in future training sessions.

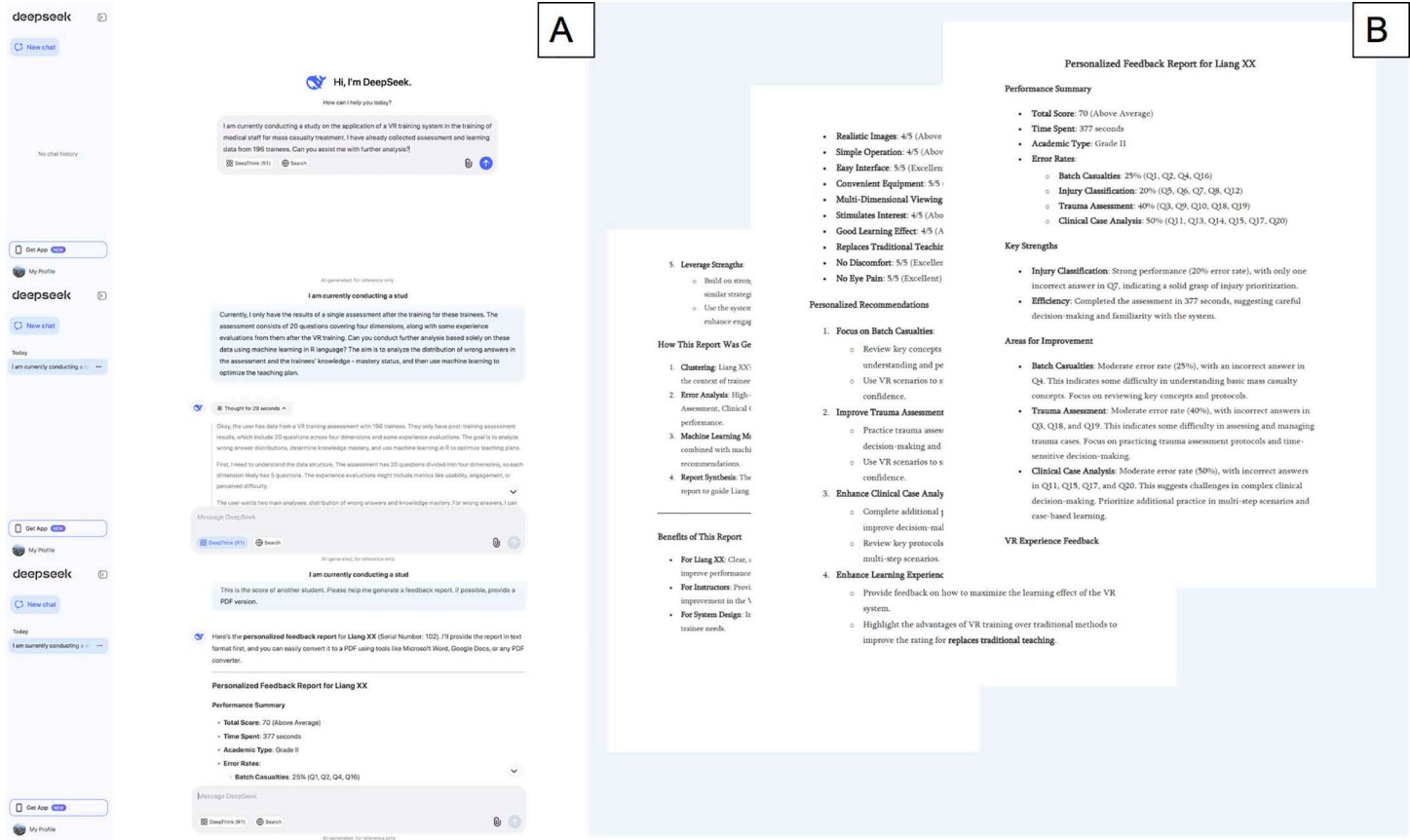

**Fig 7. AI-driven personalized feedback reports for optimizing trainee performance.** A. We input the results of VR training into the Deepseek dialogue box and wait for further suggestions; B. Based on AI, Deepseek provides feedback on trainees' performance and recommendations for future learning.

The clustering analysis revealed that the highest-performing group demonstrated notable strengths in critical decision-making during trauma assessments and clinical case analyses. These results align with existing literature emphasizing the potential of VR to enhance decision-making abilities [12], especially in high-pressure scenarios such as emergency care [13]. However, the study also found that even high-performing trainees struggled with certain aspects, particularly in clinical case analysis. This suggests a need for more contextually rich and dynamic VR scenarios that closely replicate real-world complexities. This finding contrasts with studies reporting more consistent outcomes across similar VR-based training programs [14], possibly due to differences in training complexity or learner experience.

The high error rates observed in specific training components, such as clinical case analysis, underscore the need for a more targeted approach to VR training design. Modifying VR scenarios to include more intricate, real-life cases could help address these issues. Additionally, enhancing step-by-step guidance within the VR system could provide valuable scaffolding for less experienced trainees. Findings from learner feedback emphasized that improvements in realism, scenario complexity, and interactive elements could significantly enhance both engagement and performance. These insights highlight the importance of continuous iteration in VR system design to ensure adaptability to the diverse needs of learners [14]. Furthermore, a comparison with a historical training cohort suggests improved theoretical and practical performance in learners who underwent AI-enhanced VR training. While this analysis is limited by non-randomized conditions

and variability in traditional instruction, it provides preliminary support for the effectiveness of immersive and personalized VR-based approaches.

Importantly, AI-driven feedback systems extend beyond traditional evaluation methods, offering transformative insights into both group-level instructional design and individualized learning pathways. In conventional educational settings, assessment often relies on aggregate scores derived from theoretical knowledge, skill performance, and subjective evaluations. While these metrics offer a general overview, they frequently obscure nuanced differences in learner comprehension and clinical decision-making capabilities. By contrast, AI-enhanced VR training enables detailed error tracking, behavior analysis, and response timing, which allows educators to identify widespread learning gaps—such as frequent misunderstandings in trauma triage or prioritization—and refine instructional strategies accordingly. At the individual level, AI can pinpoint a trainee's specific weaknesses based on incorrect answers and decision delays, facilitating targeted remediation and longitudinal monitoring of progress. Through this dual-layered feedback, the integration of AI significantly enhances both the precision and personalization of clinical education [15], bridging the gap between simulation-based training and real-world emergency decision-making.

Despite their advantages, implementing AI-enhanced VR systems in healthcare education remains challenging [16]. Technical limitations—including hardware demands and the need for stable system integration—can restrict deployment, particularly in resource-limited settings. Financial constraints also pose a significant barrier, especially in low- and middle-income regions where immersive technology may not be a funding priority. In addition, resistance to new technologies among faculty—especially those less familiar with AI systems—may impede adoption [17]. Effective integration therefore requires institutional commitment, robust evidence of educational value, and structured training programs to support both instructors and learners during the transition.

Virtual reality offers an immersive and interactive training experience that often surpasses traditional methods [18], particularly when augmented with AI-driven feedback. Nevertheless, some trainees may still perceive VR as a complement rather than a replacement for conventional instruction, especially when the virtual scenarios lack the perceived depth or adaptability of instructor-led teaching [19]. This study supports the potential of a hybrid instructional model that integrates VR-based simulations with traditional pedagogical approaches. Such a model could harness the experiential benefits of VR while retaining the structured guidance and contextual richness of conventional training. Future research should further explore the design and efficacy of these hybrid models to optimize learning outcomes.

### Strengths and limitations

This study provides a novel integration of machine learning techniques with immersive VR systems in the context of mass casualty training. A major strength lies in the systematic use of AI-based feedback to guide instructional design and personalized improvement strategies. Furthermore, the use of the same theoretical exam across both historical and intervention groups allows for meaningful comparison despite variability in prior training methods.

However, several limitations should be noted. First, the study was conducted in a single center with a sample size of 196 participants, which may limit generalizability. Second, the historical control group underwent non-standardized training in terms of content and duration, although the theoretical assessment remained constant. Third, the study design was cross-sectional and did not allow for long-term evaluation of skill retention. Fourth, the reliance on self-reported feedback introduces subjective bias, and the absence of physiological or real-world clinical outcome data further limits the interpretability of training efficacy.

Future research should consider multicenter studies with larger and more diverse populations to improve generalizability. Additionally, the incorporation of biometric indicators (e.g., heart rate, eye tracking) [20] and real-world clinical assessments could provide more objective and comprehensive evaluations of VR training outcomes.

### Conclusions

In conclusion, this study demonstrates the potential of integrating AI with VR training systems to optimize learning outcomes in medical education. By identifying patterns in trainee performance and tailoring instructional content to individual

needs, the combination of VR and AI provides a powerful framework for improving training efficiency and effectiveness. The findings have important implications for the design and implementation of VR-based medical training programs, particularly in high-stakes environments such as emergency medicine. Future research should continue to refine these systems to address the evolving needs of healthcare professionals.

## Supporting information

**S1 File. Scores of all participants in the theoretical examination.**
(ZIP)

## Acknowledgments

This study utilizes a shared dataset collected for VR training in mass casualty management. It should be noted that the overlapping data across our related studies is limited primarily to a small portion of the theoretical exam scores. Although these scores have been included in other research, each manuscript investigates distinct research questions, employs different analytical approaches, and presents unique findings. We have ensured transparency by specifying that the data originated from our in-house VR training system at the Emergency Department of Guangxi Zhuang Autonomous Region People's Hospital, with a sample size of 196 participants collected from January to December 2024. This clarification serves to contextualize the data reuse while underscoring the independent contributions of each study.

## Author contributions

**Conceptualization:** Zhe Li, Wan Chen, Liwen Lyu.

**Data curation:** Zhe Li, Liwen Lyu.

**Formal analysis:** Zhe Li, Xibin Xu, Liwen Lyu.

**Funding acquisition:** Zhe Li, Mingyu Pei.

**Investigation:** Zhe Li, Lei Shi, Liwen Lyu.

**Methodology:** Zhe Li, Lei Shi, Guozheng Qiu, Liwen Lyu.

**Project administration:** Zhe Li, Liwen Lyu.

**Resources:** Zhe Li, Lei Shi, Yutao Tang, Guozheng Qiu, Liwen Lyu.

**Software:** Zhe Li, Wan Chen, Liwen Lyu.

**Supervision:** Zhe Li, Guozheng Qiu, Liwen Lyu.

**Validation:** Zhe Li, Mingyu Pei, Yutao Tang, Guozheng Qiu.

**Visualization:** Mingyu Pei.

**Writing – original draft:** Zhe Li.

**Writing – review & editing:** Xibin Xu, Liwen Lyu.

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
