## [Decision Letter · Decision Letter 0]

Dear Dr. Li,

Thank you for submitting your manuscript to PLOS ONE. After careful consideration, we feel that it has merit but does not fully meet PLOS ONE’s publication criteria as it currently stands. Therefore, we invite you to submit a revised version of the manuscript that addresses the points raised during the review process.

**ACADEMIC EDITOR:** Please address all the concerns raised by reviewers 1 and 2, before I consider accepting your manuscript. 

We look forward to receiving your revised manuscript.

Kind regards,

Ziyu Qi

Academic Editor

PLOS ONE

 [Guangxi Medical and Health Appropriate Technology Development and Promotion Application Project [grant number: S2023013]�Guangxi Natural Science Foundation [grant number: 2024GXNSFAA010071] and Guangxi Natural Science Foundation [grant number: 2024GXNSFAA010012]]. 

5. We note that your Data Availability Statement is currently as follows: [All relevant data are within the manuscript and its Supporting Information files.]

Additional Editor Comments:

Please address all the concerns raised by reviewers 1 and 2, before I consider accepting your manuscript.

Reviewers' comments:

Reviewer's Responses to Questions

**Comments to the Author**

1. Is the manuscript technically sound, and do the data support the conclusions?

Reviewer #1: Yes

Reviewer #2: Yes

2. Has the statistical analysis been performed appropriately and rigorously?

Reviewer #1: Yes

Reviewer #2: I Don't Know

3. Have the authors made all data underlying the findings in their manuscript fully available?

Reviewer #1: Yes

Reviewer #2: Yes

4. Is the manuscript presented in an intelligible fashion and written in standard English?

Reviewer #1: Yes

Reviewer #2: Yes

Reviewer #1: • The manuscript does not clearly establish the theoretical foundation behind AI-enhanced VR training. Please Incorporate a stronger theoretical background, discussing educational psychology models (e.g., Kolb’s Experiential Learning Theory) and learning retention frameworks in AI-driven education.

• The study highlights the benefits of AI and VR-based training but does not provide a clear comparison with traditional training approaches. Please include a comparative analysis between VR-based and traditional teaching methods to quantify the effectiveness of AI-enhanced VR.

• The study is conducted in a single institution with 196 participants, which may limit generalizability. Please Acknowledge this limitation explicitly and suggest multicenter studies or a larger sample for future research.

• The paper extensively discusses PCA-based clustering and machine learning outputs, but it does not explicitly relate findings to real-world clinical training improvements. Please Clarify how AI-driven feedback translates into better decision-making skills in actual emergency scenarios.

• The Results section includes a data analysis section, which makes it confusing. I recommend moving all data analysis content to a separate section titled ‘Data Analysis’ to improve clarity and organization.

• The strengths and limitations of the study should be addressed in a separate section. Please add a section that discusses them separately.

• The manuscript lacks a discussion on practical challenges in implementing AI-driven VR training in healthcare settings. Please Address technological barriers, cost, and staff resistance that may hinder the integration of AI-enhanced VR training into clinical education.

Reviewer #2: The manuscript contributes to the knowledge of DeepSeek-AI in this growing space of generative AI. It is quite an interesting manuscript and should be considered for publication. I am open to further reviews.

**Do you want your identity to be public for this peer review?** For information about this choice, including consent withdrawal, please see our Privacy Policy

Reviewer #1: No

Reviewer #2: **Yes: ** OLAYINKA ADEBAJO

---

## [Author Response · Author response to Decision Letter 1]

21 Apr 2025

Dear Reviewers,

Thank you very much for your thoughtful and constructive comments. We greatly appreciate your feedback, as it has helped us clarify several aspects of the study and improve the manuscript. Below, we provide detailed responses to each of your comments:

Reviewer #1:

λ Comment :

The manuscript does not clearly establish the theoretical foundation behind AI-enhanced VR training. Please Incorporate a stronger theoretical background, discussing educational psychology models (e.g., Kolb’s Experiential Learning Theory) and learning retention frameworks in AI-driven education.

Response:

We sincerely appreciate the reviewer for their insightful observation regarding the need to strengthen the theoretical underpinnings of our study. This feedback was pivotal in identifying an area where our initial framework could be significantly enhanced. In response, we have undertaken a thorough revision of the introduction to build a more robust theoretical foundation for our AI-enhanced VR training model.

Specifically, we now explicitly integrate Kolb’s Experiential Learning Theory, meticulously mapping its four core stages—concrete experience, reflective observation, abstract conceptualization, and active experimentation—to the immersive VR scenarios, real-time AI-generated feedback, and adaptive training pathways enabled by our DeepSeek system. Additionally, we have incorporated key learning retention theories, such as the Ebbinghaus forgetting curve, to theoretically ground how AI-driven spaced repetition and personalized reinforcement mechanisms address knowledge decay over time. These pedagogical frameworks are now deeply interwoven with the technical design of our platform, demonstrating how educational psychology principles directly inform the functionality of the AI-enhanced VR training system.

For details, please refer to the revised theoretical section in lines [68–92], where these concepts are systematically linked to our methodology and research objectives. Once again, we are grateful for the reviewer’s sharp attention to this critical aspect, which has substantially improved the rigor and depth of our manuscript.

λ Comment :

The study highlights the benefits of AI and VR-based training but does not provide a clear comparison with traditional training approaches. Please include a comparative analysis between VR-based and traditional teaching methods to quantify the effectiveness of AI-enhanced VR.

Response:

Thank you for this insightful suggestion. To address this point, we have included a comparative analysis between the AI-enhanced VR training group and a historical cohort of medical trainees who received traditional training. Specifically, in Section 2.2 Participants (lines 144–149), we have clarified that all participants undertook the same standardized theoretical examination, allowing for meaningful comparison, although the traditional training group underwent non-standardized training protocols between 2021 and 2023.

In the Results section, we added a new subsection titled 3.1.2 Comparison with Historical Cohort Under Traditional Training, where we report that the VR+AI group (n=196) achieved a higher average theoretical assessment score (60.63 ± 24.68) compared to the historical traditional training group (55.79 ± 17.89). This comparison is also visualized in Figure 2, which clearly illustrates the performance difference.

Furthermore, we briefly discuss this comparison in the Discussion section (lines 386–391), noting that while the training methods in the historical cohort were not standardized, the consistency of the examination content allows for an informative preliminary comparison that supports the improved learning outcomes facilitated by AI-enhanced VR training.

λ Comment :

The study is conducted in a single institution with 196 participants, which may limit generalizability. Please Acknowledge this limitation explicitly and suggest multicenter studies or a larger sample for future research.

Response:

We thank the reviewer for this insightful comment. We fully acknowledge this limitation and have now explicitly addressed it in the revised Discussion (paragraph 7). As the study was conducted in a single institution with 196 participants, we recognize that the findings may not be generalizable to other educational settings or learner populations. We agree that future research should aim to include larger, more diverse samples across multiple centers to enhance external validity. Identifying this limitation also helps us define the next steps in our research trajectory—by expanding the scale of implementation, we will be able to conduct comparative analyses across institutions and learner subgroups, which will further validate and refine our AI-enhanced VR training model.

λ Comment :

The paper extensively discusses PCA-based clustering and machine learning outputs, but it does not explicitly relate findings to real-world clinical training improvements. Please clarify how AI-driven feedback translates into better decision-making skills in actual emergency scenarios.

Response:

We sincerely appreciate the reviewer’s thoughtful observation. In the revised Discussion (paragraph 4), we have elaborated on how AI-driven feedback contributes to real-world clinical training. Specifically, we explain that individualized error analysis and pattern recognition can help learners identify critical decision-making weaknesses—such as delayed triage or misclassification of injury severity—and allow instructors to target these gaps in follow-up training. While our current study did not directly measure real-world clinical performance, we believe this AI-enhanced feedback loop holds strong potential to improve emergency response readiness through more personalized and adaptive training.

λ Comment :

The Results section includes a data analysis section, which makes it confusing. I recommend moving all data analysis content to a separate section titled ‘Data Analysis’ to improve clarity and organization.

Response:

Thank you for your valuable comment. In response, we have integrated the relevant technical and methodological details—such as machine learning model construction, clustering, and PCA—into Section 2.5 (now renamed “Data Collection and Analytical Methods”) to avoid redundancy. Correspondingly, we streamlined the Results section (Sections 3.2 to 3.5), removing or condensing repeated descriptions of methods and focusing on key findings. We believe this revision improves the clarity and structure of the manuscript.

λ Comment :

The strengths and limitations of the study should be addressed in a separate section. Please add a section that discusses them separately.

Response:

Thank you for your helpful suggestion. We have revised the manuscript to include a new subsection titled “Strengths and Limitations” at the end of the Discussion section . In this section, we summarize the key strengths of our study, including the novel integration of VR simulation and machine learning–based feedback, the application of unsupervised clustering to reveal learner patterns, and the use of a real-world emergency training scenario grounded in mass casualty triage. These aspects contribute to the originality and applied value of our work in the domain of medical education.

At the same time, we acknowledge several limitations. The study was conducted at a single institution with a relatively limited sample size, which may affect the generalizability of the findings. Furthermore, although our clustering and modeling approaches provided insight into learning performance and feedback pathways, we did not track long-term knowledge retention or real-world clinical behavior changes. Finally, while the machine learning models were interpretable to some extent, the integration of model outputs into dynamic instructional redesign still requires further validation in prospective studies. These limitations are discussed to provide a balanced and transparent assessment of our study’s implications.

λ Comment :

The manuscript lacks a discussion on practical challenges in implementing AI-driven VR training in healthcare settings. Please Address technological barriers, cost, and staff resistance that may hinder the integration of AI-enhanced VR training into clinical education.

Response:

We sincerely thank the reviewer for raising this important and practical concern. In the revised Discussion (paragraph 5), we have added a paragraph specifically addressing the real-world challenges of implementing AI-enhanced VR training in healthcare settings. These challenges include the high initial cost of acquiring VR equipment and integrating AI technologies, the demand for ongoing technical support and maintenance, and the need for infrastructure upgrades in some clinical institutions. Additionally, staff resistance due to unfamiliarity with immersive technologies and concerns about disrupting traditional teaching routines may hinder adoption. Recognizing these barriers is crucial for developing feasible implementation strategies, and future research should incorporate stakeholder engagement, cost-effectiveness analysis, and implementation science frameworks to facilitate sustainable integration into clinical education systems.

Reviewer #2:

λ Comment :

The manuscript contributes to the knowledge of DeepSeek-AI in this growing space of generative AI. It is quite an interesting manuscript and should be considered for publication. I am open to further reviews.

Response:

We sincerely thank the reviewer for the positive and encouraging feedback. We are honored that you found our manuscript interesting and valuable in the context of the rapidly evolving field of generative AI. As researchers, we are equally excited about the transformative potential of AI technologies like DeepSeek in enhancing medical education, and we are grateful for the opportunity to contribute to this important and growing area of study.

---

## [Editor Report · Decision Letter 1]

DeepSeek-AI-Enhanced Virtual Reality Training for Mass Casualty Management: Leveraging Machine Learning for Personalized Instructional Optimization

PONE-D-25-11072R1

Dear Dr. Li,

We’re pleased to inform you that your manuscript has been judged scientifically suitable for publication and will be formally accepted for publication once it meets all outstanding technical requirements.

Kind regards,

Ziyu Qi

Academic Editor

PLOS ONE
---

## [Editor Report · Acceptance letter]

PONE-D-25-11072R1

PLOS ONE

Dear Dr. Li,

I'm pleased to inform you that your manuscript has been deemed suitable for publication in PLOS ONE. Congratulations! Your manuscript is now being handed over to our production team.

Kind regards,

on behalf of

Mr. Ziyu Qi

Academic Editor

PLOS ONE